# Evaluating Two Educational Interventions for Enhancing COVID-19 Knowledge and Attitudes in a Sample American Indian/Alaska Native Population

**DOI:** 10.3390/vaccines12070787

**Published:** 2024-07-17

**Authors:** Maya Asami Takagi, Simone T. Rhodes, Jun Hwan Kim, Maxwell King, Stephanie Soukar, Chad Martin, Angela Sasaki Cole, Arlene Chan, Ciara Brennan, Stephen Zyzanski, Barry Kissoondial, Neli Ragina

**Affiliations:** 1College of Medicine, Central Michigan University, Mount Pleasant, MI 48858, USA; rhode1s@cmich.edu (S.T.R.); kim16j@cmich.edu (J.H.K.); king5mj@cmich.edu (M.K.); souka1sl@cmich.edu (S.S.); marti9c@cmich.edu (C.M.); sasak1am@cmich.edu (A.S.C.); chan3a@cmich.edu (A.C.); brenn2ce@cmich.edu (C.B.); ragin1n@cmich.edu (N.R.); 2Department of Internal Medicine, University of California Davis School of Medicine, Sacramento, CA 95817, USA; 3Department of Family Medicine and Community Health, School of Medicine, Case Western Reserve University, Cleveland, OH 44106, USA; sjz@case.edu; 4College of Medicine Affiliated Community Clinic, Central Michigan University, Mount Pleasant, MI 48858, USA; bkissoondial@sagchip.org

**Keywords:** American Indian/Alaska Native populations, COVID-19 vaccine hesitancy, educational interventions, health disparities, health equity

## Abstract

Background: The COVID-19 pandemic has exacerbated existing healthcare disparities among American Indian/Alaska Native (AI/AN) populations rooted in historical traumas and systemic marginalization. Methods: This study conducted at a single Indian Health Service (IHS) clinic in central Michigan evaluates two educational interventions for enhancing COVID-19 knowledge and attitudes in a sample AI/AN population. Utilizing a pre/post-intervention prospective study design, participants received either a video or infographic educational intervention, followed by a survey assessing their COVID-19 knowledge and attitudes. Results: The results indicate significant improvements in knowledge and attitudes post-intervention, with both modalities proving effective. However, specific factors such as gender, political affiliation, and place of residence influenced COVID-19 attitudes and knowledge, emphasizing the importance of tailored interventions. Conclusions: Despite limitations, this study highlights the critical role of educational interventions in addressing vaccine hesitancy and promoting health equity within AI/AN communities. Moving forward, comprehensive strategies involving increased Indian Health Service funding, culturally relevant interventions, and policy advocacy are crucial in mitigating healthcare disparities and promoting health equity within AI/AN communities.

## 1. Introduction

American Indian/Alaska Native/Native American (AI/AN) populations have long experienced disparities in healthcare access and outcomes rooted in a history of colonization, forced relocation, and systemic marginalization [1,2,3,4]. These historical traumas have contributed to persistent health inequities, making AI/AN populations particularly vulnerable to the devastating effects of the COVID-19 virus [5,6,7,8,9,10,11]. The COVID-19 pandemic has presented challenges throughout the world. Among the various sectors and populations that the pandemic has affected, AI/AN communities have faced unique and disproportionate burdens stemming from historical, socioeconomic, and health-related disparities. The COVID-19 pandemic has disproportionately affected the AI/AN population across the country. AI/AN populations endured (i) COVID-19 infection rates over 3.5 times higher than those in non-Hispanic whites, (ii) hospitalization rates four times higher compared to non-Hispanic whites, and (iii) higher rates of mortality at younger ages than non-Hispanic whites [12,13]. Additionally, a number of studies have demonstrated greater COVID-19 vaccine hesitancy amongst minority groups compared to whites in the United States [14,15]. These disparities highlight the urgent need for targeted interventions within these communities.

Vaccination against COVID-19 represents a critical tool in mitigating the spread of the virus and protecting vulnerable populations [16]. At the end of 2021, the vaccination rate of American Indians in Michigan was 52%, below the ideal 70% vaccination rate that was necessary to protect the community from disease [17]. However, disparities in vaccination uptake have emerged, raising concerns about equitable access and vaccine acceptance among AI/AN populations. 

Educational interventions play a pivotal role in addressing these challenges, offering opportunities to dispel myths and educate communities to make informed decisions about vaccination. By examining the attitudes and beliefs among AI/AN populations in the context of the COVID-19 pandemic, this study aims to contribute to the understanding of vaccine hesitancy and the effectiveness of educational interventions in promoting vaccination acceptance within this marginalized population. 

The current literature highlights how misinformation can exacerbate health disparities by complicating decision-making processes and undermining trust in healthcare institutions. In addition, additional studies emphasize the significance of establishing and sustaining trust between patients and healthcare providers as a critical strategy to mitigate the impact of misinformation and promote health equity [18,19]. These insights demonstrate the complexities of health communication in addressing public health challenges and emphasize the need for culturally responsive interventions that foster trust and engagement within AI/AN communities. By integrating these perspectives, this study not only seeks to advance our knowledge on the COVID-19 virus and vaccines among AI/AN populations, but also informs strategies to enhance healthcare delivery and promote health equity in marginalized populations.

## 2. Materials and Methods

### 2.1. Study Design

A pre/post-intervention prospective study design was used to collect data using questionnaires at a single outpatient Indian Health Service (IHS) clinic in central Michigan. This study was conducted from 22 February 2022 to 4 January 2024. Research assistants recruited patients in the waiting room of the selected clinic to complete the questionnaire. The study design included a pre-survey followed by a randomized educational intervention delivered as either a video or infographic, and concluded with a post-survey (Figure 1). Upon obtaining informed consent, participants completed a questionnaire using a pre-installed survey on an iPad provided by the research team or via their personal smartphone devices by scanning a QR code. Subsequently, participants’ attitudes and knowledge regarding COVID-19 and its vaccines were evaluated to assess the effectiveness of the interventions. All data collected in this study were anonymized, with participant email addresses stored separately in a secure document as the sole identifying information. Questionnaires and educational materials were administered by CITI-trained research assistants from the Central Michigan University (CMU) College of Medicine. Approval and supervision were provided by the CMU College of Medicine Research Institutional Review Board (IRB) and the central Michigan AI tribe to ensure adherence to ethical standards and participant confidentiality. 

### 2.2. Participants

Participants were recruited from a single Indian Health Service outpatient clinic in rural Michigan which served 4123 active patients between 2018 and 2021. The inclusion criteria specified patients from this clinic who were over 18 years old and could understand English. The exclusion criteria encompassed individuals who were not clinic patients, those under 18 years old, or those who could not understand English.

### 2.3. Questionnaires and Educational Interventions

The study protocol questionnaire included a validated 56-item pre-survey, a randomized educational intervention, and a 38-item post-survey (Figure 1), which were gathered using Qualtrics Online Survey Platform between 22 February 2022 and 4 January 2024 [20] (Appendix A). These three components were intended to be completed during a single session, though participants had the flexibility to pause and resume as needed, such as when waiting for their appointment. The pre-survey collected data on demographics, COVID-19 virus and vaccine knowledge, COVID-19 vaccination status, and beliefs and concerns about the COVID-19 vaccines. Demographic, virus and vaccine knowledge, and vaccination status questions were multiple-choice, with all questions being optional. Attitudes toward COVID-19 and its vaccine were assessed using 29 items on a three-point Likert scale: 2 for agree, 1 for unsure, and 0 for disagree. There were also seven multiple-choice questions to evaluate knowledge about COVID-19 and vaccines. Participants were then randomly assigned via Qualtrics to either a seven-minute educational video about COVID-19 and its vaccines or an educational infographic about COVID-19 and its vaccines (Appendix A). Those assigned the infographic had to spend at least four minutes reviewing the document before proceeding. Both educational materials were based on information from the Centers for Disease Control and Prevention (CDC) and the World Health Organization (WHO). After the educational intervention, participants answered the same COVID-19 knowledge and attitude questions as in the pre-survey, excluding demographic questions. Upon completing the post-survey, participants received a USD 10 gift card as a reward for their participation in all three components.

Knowledge levels were assessed through inquiries covering various aspects, such as measures to protect against COVID-19 transmission, modes of COVID-19 spread, vaccine effects, COVID-19 vaccine efficacy, post-vaccine behavior, COVID-19 virus side effects, and COVID-19 vaccine development. Attitudes were evaluated across multiple dimensions, including trust in COVID-19 vaccine effectiveness and safety, personal belief in the benefits of COVID-19 vaccination, confidence in COVID-19 vaccine testing and results, their perception of rapid COVID-19 vaccine development, concerns about COVID-19 vaccine side effects, concerns about long-term effects of COVID-19 vaccines, challenges in discerning trustworthy vaccine information, the influence of trusted sources on vaccination, concerns about missing work due to vaccine side effects, etc.

### 2.4. Study Sample and Statistical Analysis

This Indian Health Service clinic served 4123 active patients between 2018 and 2021, prior to the start of this study. The sample was randomly selected from individuals in the clinic’s waiting room during the study period. The sample size was determined based on a desire to detect a medium intervention effect size (half of a standard deviation difference between means) with a power of 95% and an alpha value of 0.05, resulting in an estimated sample size of 210.

Changes in attitudes and knowledge were assessed using paired *t*-tests to compare pre-intervention and post-intervention survey item responses. Selected pre/post-intervention analyses were also computed using non-parametric statistics (Wilcoxon matched-pairs signed-rank test) with comparable results. Additionally, independent *t*-tests were used to compare changes in attitude and knowledge items between participants exposed to the infographic versus videographic intervention. The influence of demographic factors on changes in attitudes and knowledge was explored through independent *t*-tests and one-way ANOVAs. The assumption of homogeneity of variance was assessed for both t-tests and ANOVAs. Due to the large number of tests computed for item change, the Benjamini–Hochberg procedure was applied to adjust for multiple hypothesis testing. The results are reported for each individual item tested as computed and then compared with an adjusted p-value for comparison. 

## 3. Results

The 273 participants represent 54% of invited participants and 6.6% of the study population. Participants from 27 different counties across Michigan completed the pre-survey, the educational intervention, and the post-survey (Figure 2). The demographic profile of these participants is detailed in Table 1. Notably, the cohort comprised predominantly females (67%), with the most common age brackets being 25–34 years (22%) and 35–44 years (21%). The majority identified as non-Hispanic (91%), and a significant portion resided in micropolitan areas (79%), as classified by Core Based Statistical Areas. Regarding political affiliations, most participants identified as democratic (30%) or with alternative affiliations (31%). In terms of COVID-19 impacts, more than half of the participants (55%) reported pandemic-related disruptions to their employment status. Additionally, 51% of participants or someone in their household identified as essential workers during the pandemic. The majority (88%) self-tested for COVID-19, with 90% of participants reporting a history of positive test results. Over half (58%) of the participants knew someone hospitalized or deceased due to COVID-19. Regarding co-morbidities, 47% reported having at least one at-risk medical condition, while 33% reported having two or more. Concerning household members, 40% indicated one household member with an at-risk condition, and 36% indicated two or more. Participants were also queried about adherence to a list of CDC guidelines, such as mask wearing, social distancing, and hand hygiene, with the majority (74%) reporting compliance with four or more precautions. Regarding seasonal vaccinations, 58% received the influenza vaccine last year, and 52% either received or planned to receive it this year. Lastly, 59% of this study cohort were already partially or completely vaccinated.

Analysis of COVID-19 virus and vaccine knowledge before and after the educational intervention indicated that knowledge about five out of the seven knowledge-based questions improved (Table 2). After the intervention, participants demonstrated increased knowledge about protection against and a reduction in COVID-19 transmission (*p* < 0.001), how vaccines work (*p* = 0.008), how the COVID-19 vaccines work (*p* < 0.001), COVID-19 vaccine side effects (*p* = 0.013), and COVID-19 vaccine development (*p* < 0.001). Topics regarding how COVID-19 spreads (*p* = 0.94) and being cautious in the public (*p* = 0.138) did not show significant change. Of the knowledge items that showed overall statistical significance, when stratifying the participants into various demographic groups (Appendix A), those who identified as female improved their knowledge regarding post-vaccine behaviors compared to those who identified as male (*p* = 0.014) and those who identified as a political affiliation other than republican, democratic, or independent improved their knowledge regarding COVID-19 vaccine development (*p* = 0.016). Belief in COVID-19 vaccine efficacy (*p* = 0.029) was most improved in those identified as republican compared to other political affiliations; and individuals whose employment status was affected by the pandemic also improved their knowledge about COVID-19 vaccine efficacy (*p* = 0.043) compared to those whose employment status was not affected by the pandemic. No other demographic associations were seen among other knowledge items (Appendix A). 

Our data demonstrated that overall, 7 out of 29 attitude items showed statistically significant changes, and 2 out of 29 items showed meaningful trends. Our findings indicated that participants showed an increase in their trust in the vaccine (*p* < 0.001) and increased their belief that the COVID-19 vaccine would benefit their body (*p* = 0.012) (Table 3). More had confidence that the COVID-19 vaccine underwent adequate testing (*p* = 0.012), and more cited a source that they trust that told them to receive a COVID-19 vaccine (*p* = 0.006). The results also showed a significant reduction in negative misconceptions, such as less individuals believing that the vaccine was developed too quickly (*p* = 0.016), as well as reduced concerns about the side effects of the vaccine (*p* = 0.01) and the long-term effects of the vaccine (*p* < 0.001). Trends were observed in decreased concerns about knowing whom to trust for COVID-19 information (*p* = 0.057) and fewer participants being worried about missing work to receive this vaccine (*p* = 0.052). There was no statistical significance or meaningful trends in the other 21 attitude topics, such as concern about contracting the virus, belief in CDC recommendations, etc. (Appendix A). Of the attitude items that showed overall statistical significance, when stratifying the participants into various demographic groups, those who identified as male (*p* = 0.021) and individuals who resided in metropolitan areas (*p* = 0.014) showed a greater decrease in concerns about COVID-19 vaccine side effects (Appendix A). Also, those who identified as male (*p* = 0.021), individuals who had never tested themselves for COVID-19 (*p* = 0.038), and individuals with two or more co-morbidities (*p* = 0.002) showed a greater decrease in concerns about the long-term effects of the COVID-19 vaccines. Individuals with no co-morbidities increased their belief in the benefits of the COVID-19 vaccine (*p* = 0.04); fewer individuals with two or more co-morbidities believed that the COVID-19 vaccine was developed too quickly (*p* = 0.01); and the more CDC precautions the individual followed, the more they were concerned about missing work due to COVID-19 vaccine side effects (*p* = 0.02). No other demographic associations were seen among other attitude items (Appendix A).

The significant knowledge and attitude item changes by the participants’ vaccination status shows that individuals who were unvaccinated and hesitant regarding COVID-19 vaccinations increased their knowledge the most in topics regarding vaccine effects (*p* = 0.004) and COVID-19 vaccine efficacy (*p* < 0.001) (Table 4). In addition, both unvaccinated groups increased their beliefs in the benefits of COVID-19 vaccinations after the intervention (*p* = 0.011). No other demographic associations were seen among other knowledge and attitude items (Appendix A).

However, those who received the infographic educational intervention were less concerned about the COVID-19 vaccine (*p* = 0.036), and the more CDC precautions an individual followed, the more likely they were to receive the COVID-19 vaccine (*p* = 0.03) (Appendix A). 

## 4. Discussion

### 4.1. The Impact and Role of Educational Interventions in Addressing Public Health Issues among Populations with Disproportionate Outcomes

Overall, the educational interventions have demonstrated positive outcomes, enhancing COVID-19 knowledge and attitudes while reducing misconceptions about vaccinations. Similar to the results of this study, Takagi et al. (2023) demonstrated how the same educational interventions increased knowledge amongst patients across 18 Michigan counties [20]. A minor difference was observed between the video and infographic regarding concerns about the COVID-19 vaccine. This discrepancy may highlight the importance of cultural relevance in educational materials. For instance, the inclusion of a non-AI/AN infectious disease physician in the video may not have resonated as strongly with AI/AN communities as culturally representative personnel would. Integrating cultural components into educational interventions can foster greater trust and engagement, ultimately enhancing their impact within AI/AN communities.

At the onset of the pandemic, AI/AN communities faced notably low vaccination rates. However, by April 2021, they had become the most vaccinated minority group, reflecting successful efforts by tribal leadership to tailor vaccine distribution strategies to their community’s needs and leveraging existing trusted resources [21,22,23]. Initiatives such as the IHS’s COVID-19 Vaccine Task Force (VTF) and the involvement of native language speakers from the Cherokee and Navajo Nations played pivotal roles in educating community members about COVID-19 vaccine safety and efficacy [24,25]. Despite these achievements, vaccine hesitancy persists, influenced in part by historical experiences of infectious diseases like smallpox, tuberculosis, measles, and diphtheria since European contact in the 15th century which have contributed to ongoing health disparities and mistrust within AI/AN communities [11]. Various strategies are available for promoting uptake and decreasing vaccine hesitancy, with educational interventions, like the ones utilized in this study, serving as an additional tool in the toolbox alongside other resources such as native language speakers and community health educators. Initiatives such as the Diné (Navajo) Teachings and Public Health Students Informing Peers and Relatives about Vaccine Education (RAVE) intervention aim to address this hesitancy by serving as trusted messengers, delivering culturally relevant vaccine education [26]. Leveraging the ethnic concordance of providers and patients, along with community health educators and Traditional Knowledge Holders utilizing the Hózhó Resilience Model, further enhances trust and empathy in healthcare interactions [27,28,29].

Derived from Warne and Lajimodiere, Figure 3A summarizes the cumulative effects of historical trauma alongside gestational, childhood, and adulthood stressors on chronic disease disparities amongst AI/AN populations [30]. Figure 3B illustrates how vaccination and educational interventions counteract and reduce these disparities. Educational interventions represent a valuable tool in addressing public health challenges, providing targeted strategies to enhance health literacy, promote preventive healthcare behaviors, and address concerns surrounding factors contributing to chronic disease disparities. These interventions are often guided by theoretical frameworks, such as the Health Belief Model. This model emphasizes individual perceptions of susceptibility, severity, benefits, and barriers to health actions, helping tailor interventions to the specific needs and beliefs of AI/AN communities [31]. 

In assessing the impact of educational interventions, future research could benefit from incorporating measures of behavioral intention alongside knowledge and perception outcomes. While this study focused primarily on knowledge and attitude changes, the inclusion of behavioral intention measures, such as engagement in disease prevention activities, could provide deeper insights into the practical impacts of these interventions on health behaviors within AI/AN populations. This comprehensive approach would help elucidate how educational efforts translate into actionable health decisions and contribute to ongoing efforts to improve vaccination rates and reduce health disparities among AI/AN communities.

By offering culturally sensitive and appropriate health education, these interventions empower individuals to make informed decisions about their health, advocate their healthcare needs, and engage in preventative health measures, such as vaccination [32,33,34,35]. For example, previous culturally tailored interventions in smoking cessation have shown to be impactful among AI/AN populations [36,37]. Furthermore, educational initiatives have the potential to foster trust and collaboration between healthcare providers and AI/AN communities, facilitating more effective healthcare delivery and improving outcomes [38,39]. Through comprehensive educational efforts tailored to the historical and cultural contexts of AI/AN populations, these initiatives can contribute to reducing chronic disease disparities and promoting holistic well-being across generations.

### 4.2. Factors Influencing COVID-19 Attitudes and Knowledge: Gender, Political Affiliation, Place of Residence, Number of CDC Precautions Followed, and Number of Co-Morbidities

#### 4.2.1. Gender

This study identified females as having greater concerns about challenges in accessing vaccination sites and arranging care for dependents to receive the vaccine, as well as showing a greater improvement in knowledge about post-vaccine behaviors after the educational intervention. However, there was no significant change in vaccination status compared to males. Although previous studies have found that females and parents were more likely to avoid the COVID-19 vaccine, this study showed no statistically significant differences in vaccination status when stratifying by gender [40,41]. The studies which demonstrate females having lower COVID-19 vaccination rates are correlated with potential side effects of the vaccination. Many studies have demonstrated that biological females have a more robust immune response to vaccination and, subsequently, stronger side effects compared to biological males [42,43,44]. In turn, these side effects may compromise the parents’ ability to care for their dependents [40,45]. The differences in concern about vaccine side effects can also be attributed to societal factors, such as historical medical mistrust and a lack of female representation in the healthcare system [45,46]. The rates of medical mistrust and vaccination hesitancy have only increased following the COVID-19 pandemic [47,48]. While this study did not show statistically significant differences amongst vaccination status, addressing these biological and societal concerns regarding vaccination side effects during educational interventions can help build trust amongst female patients.

#### 4.2.2. Political Affiliation

According to the American Research Collaborative, Native Americans predominantly voted for the democratic party over the republican party [49]. Our study found that those who identified as republican improved their knowledge regarding COVID-19 vaccine efficacy the most compared to those with other political affiliations after the educational intervention. In contrast, those who identified with a political affiliation other than republican, democratic, or independent demonstrated significant improvements in their knowledge regarding COVID-19 vaccine development. The politicization of the COVID-19 pandemic played a substantial role in shaping public perception regarding vaccination. While our study did not reveal any differences in attitudes or vaccination status amongst political affiliations, several studies have shown that those identifying with republican ideologies are less likely to be vaccinated against COVID-19, have higher rates of vaccine hesitancy, and are less amenable to other public health efforts, such as masking, compared to other political affiliations [15,50,51,52]. In addition, a study in France found that those who voted for far-left or far-right political candidates, or those who did not vote at all, were significantly more likely to refuse the vaccine [53]. The relationship between political affiliations, public health behaviors, and vaccine acceptance is complex and highlights the need for targeted interventions and inclusive messages to ensure widespread vaccine uptake and mitigate the impact of political polarization in public health initiatives.

#### 4.2.3. Place of Residence

Population density affects access to healthcare resources, which in turn impacts vaccination status. Studies have demonstrated disparities in vaccination among the AI/AN communities, with individuals residing in rural areas facing additional accessibility challenges [12,54]. Although our study did not show differences in vaccination status in micropolitan versus metropolitan areas, this study did find that individuals who resided in metropolitan areas were less concerned about COVID-19 vaccine side effects. Studies have shown that metropolitan areas are associated with higher vaccination rates compared to micropolitan areas due to variables, such as better healthcare infrastructure, accessibility to vaccination sites, and differences in socioeconomic status and education levels [55,56]. Recognizing these unique challenges is critical in implementing effective interventions in rural and micropolitan areas.

#### 4.2.4. Number of CDC Precautions Followed

Throughout the pandemic, the CDC released various precautions to help the public limit the spread of the virus, such as social distancing and handwashing. Prior studies have shown that those who had never smoked before compared to former smokers were significantly less adherent to masking [57]. Native Americans are at a 32% higher risk of smoking or being exposed to smoke compared to other ethnic groups, suggesting a potential association between smoking habits and adherence to preventive measures [58]. The CDC also endorsed handwashing as a preventative measure, which has also been shown to be a common precaution practiced by adults across the United States [59]. Our study also revealed that handwashing was the most reported precaution amongst our participants. In addition, our study revealed that individuals who followed more precautions were more likely to be worried about missing work due to vaccine side effects and were more likely to get vaccinated. Their commitment to preventive measures may suggest a proactive approach to health protection for both them and others. In addition, individuals may perceive potential vaccination side effects as a hindrance to their daily responsibilities, such as missing work. One study demonstrated the risk of staff shortages due to the inability to work after COVID-19 vaccine boosters [60]. This correlation between adherence to more precautions and missing work may reflect more broadly on the balance of personal well-being, general public health, and professional commitments—additional factors to consider when implementing public health initiatives.

#### 4.2.5. Number of Co-Morbidities

While our study revealed only one significant change in knowledge in relation to the number of co-morbidities, individuals without co-morbidities acknowledged more personal benefits of vaccination compared to those with co-morbidities. Furthermore, individuals with two or more co-morbidities exhibited decreased negative perceptions of COVID-19 compared to those with one or no co-morbidities. Given that AI/AN communities exhibit lower life expectancy than most other ethnic groups, partly due to an increased likelihood of obesity and diabetes, addressing misconceptions and promoting positive attitudes toward COVID-19 vaccination become imperative [35,61,62]. According to the Great Lakes Inter-Tribal Epidemiology Center 2021 report of AI/AN communities in Michigan, Minnesota, and Wisconsin, AI/AN individuals reported having a higher prevalence of hypertension, at 35.7%, and of diabetes, at 16%, compared to their white counterparts in this three-state area [63]. During the peak of the pandemic, pre-existing co-morbidities, such as obesity and diabetes, were reported as factors that led to more severe COVID-19 outcomes [64,65]. Therefore, targeted interventions that address concerns and promote vaccination uptake, especially among individuals with co-morbidities, can play a pivotal role in reducing the disproportionate impact of COVID-19 on AI/AN communities.

### 4.3. Limitations

The logistical challenges encountered during this study led to a prolonged data collection timeline. Throughout this period, various pandemic-related factors, such as the emergence of new COVID-19 strain variants, fluctuations in COVID-19 cases, the introduction of booster shot recommendations, and changes in recommendations by public health leaders, potentially impacted the participants’ opinions over time. Considering the pre–post design of our study, it is important to note that the initial pre-test may have conditioned participants’ responses to the intervention, influencing their subsequent perceptions and attitudes. While including a no pre-test control group was beyond the scope of this study, future research could benefit from incorporating such controls to better elucidate the true impact of the intervention. This would help differentiate between the effects of the intervention itself and any potential biases introduced by the study design.

Although efforts were made to ensure demographic diversity, the study population consisted primarily of patients from a single IHS center who did not drop out, accepted participation, met the inclusion criteria, and were in the clinic’s waiting room, potentially introducing selection bias and limiting generalizability. The surveys were distributed at the clinic, which excludes individuals facing significant transportation barriers and those who may not be eligible for IHSs. Additionally, the surveys required individuals to be literate in English and were administered on either an iPad or smartphone device, further limiting this study’s inclusivity by excluding non-English-speaking individuals and those with limited proficiency in technology.

Furthermore, while the educational intervention was approved by the tribal council, it did not incorporate any cultural components, potentially limiting its effectiveness within AI/AN communities. It is also worth noting that the clinic conducted extensive COVID-19 vaccination efforts, including setting up clinics at multiple locations, working with the local health department, and conducting targeted outreach and publicity campaigns. These comprehensive efforts to promote vaccination, including walk-in clinics and public relations activities, may have influenced the study population’s attitudes and behaviors regarding COVID-19 vaccination independently of our educational intervention. Despite these limitations, the present study addresses several gaps in previous research, including the longitudinal assessment of intervention effects and a comparison of different educational modalities.

## 5. Conclusions

This study conducted at a single IHS clinic in central Michigan illustrates the critical role of educational interventions in addressing vaccine hesitancy among AI/AN populations, especially amidst the challenges posed by the COVID-19 pandemic.

One key finding of the present study is the comparable effectiveness of both the video and infographic interventions in increasing COVID-19 knowledge among participants. While these results highlight the potential of different modalities, further investigation is needed to determine whether the observed outcomes were primarily influenced by the content or format of the educational materials. Future studies could explore this by comparing various types of educational content, including less well-designed materials or no intervention at all, to better understand optimal strategies for promoting health literacy and behavior change within these communities. This suggests that investments in either modality could yield positive outcomes. However, it is crucial to consider potential barriers for individuals associated with each intervention, such as hearing or vision impairments for videos and reading impairments for infographics. Addressing these barriers through accessible formats or alternative modalities can enhance the inclusivity and effectiveness of educational interventions.

Furthermore, the historical underfunding of IHSs underscores the urgent need for increased financial support and allocation of resources toward interventions tailored to AI/AN populations [66,67]. Investing in educational initiatives, community educators, native language speakers, and Traditional Knowledge Holders can address healthcare disparities and promote health equity within these communities. By bolstering funding from culturally sensitive interventions, policymakers and healthcare leaders can take proactive steps toward improving healthcare access and outcomes for AI/AN populations.

While this study provides valuable insights into the effectiveness of educational interventions, it also highlights the importance of addressing barriers, integrating cultural components, and advocating increased funding to ensure equitable healthcare outcomes.

## Figures and Tables

**Figure 1 vaccines-12-00787-f001:**
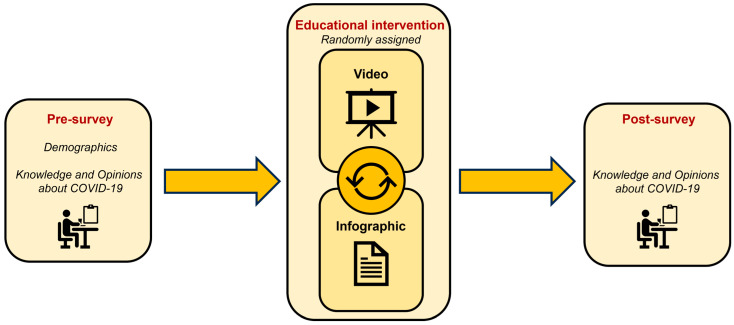
Study design workflow. Participants completed an initial validated pre-survey covering demographics, 7 COVID-19 knowledge topics, and 29 vaccine attitude topics [20]. They were then randomly assigned to a video or infographic intervention. Finally, participants completed a post-survey covering 7 COVID-19 knowledge topics and 29 vaccine attitude topics.

**Figure 2 vaccines-12-00787-f002:**
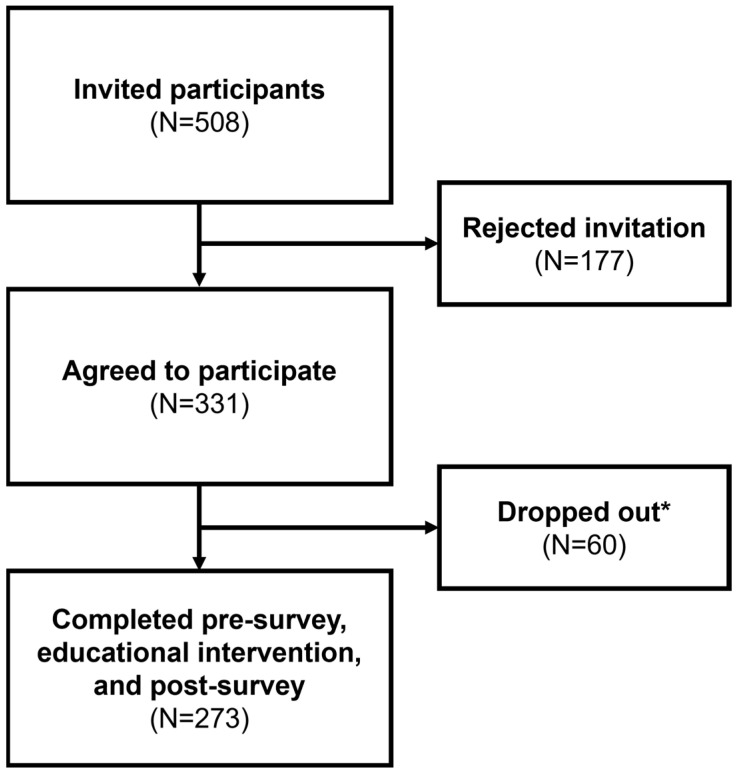
A flow diagram of the response and completion rate. * Participants who dropped out include individuals who started the survey but did not complete it.

**Figure 3 vaccines-12-00787-f003:**
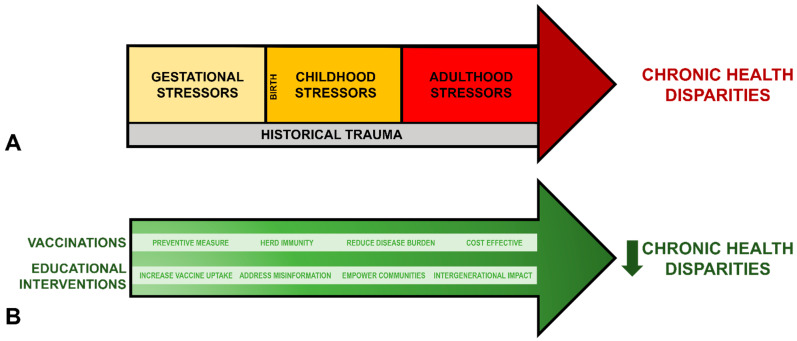
The proposed model for the basis of chronic health disparities among the AI/AN population and the role of vaccinations and educational interventions in chronic health disparities. (**A**) The cumulative effects of historical trauma and intergenerational stressors on chronic disease disparities among the AI/AN population. The historical trauma encompasses centuries of colonization, forced assimilation policies, and cultural genocide [5,6,7,8,9,10,11]. This trauma is rooted in the systemic displacement of AI/AN communities, a loss of land, language, traditional practices, as well as the experiences within AI boarding schools. Gestational stressors include maternal exposure to chronic stress, environmental toxins, and inadequate prenatal care. Childhood stressors include the impact of witnessing and experiencing abuse and violence, enduring food and family insecurity, and facing the disruption of home culture and community cohesion. Finally, adulthood stressors include high rates of alcoholism, persistent poverty, limited access to nutritious foods, and experiences of racism and discrimination. Collectively, these experiences exacerbate the burden of chronic disease within AI/AN communities, perpetuating a cycle of health disparities across generations [30]. (**B**) Vaccinations and educational interventions are integral in reducing chronic health disparities by preventing infectious diseases and addressing underlying factors contributing to poor health outcomes.

**Table 1 vaccines-12-00787-t001:** Overview of study cohort demographics (N = 273).

Characteristic	Total N (%)
Gender	
Female	183 (67%)
Male	85 (31%)
Other (transgender male, gender variant/non-conforming, prefer not to say)	5 (2%)
Age	
18–24 years old	39 (14%)
25–34 years old	60 (22%)
35–44 years old	57 (21%)
45–54 years old	45 (16%)
55–64 years old	44 (16%)
65+ years old	28 (10%)
Ethnicity	
Hispanic	22 (9%)
Non-Hispanic	224 (91%)
Residence	
Nonmetropolitan	21 (8%)
Micropolitan	216 (79%)
Metropolitan	35 (13%)
Political affiliation	
Republican	40 (15%)
Democrat	80 (30%)
Independent	65 (24%)
Something else	84 (31%)
Employment status	
Affected by COVID-19 pandemic	128 (55%)
Not affected by COVID-19 pandemic	106 (45%)
Essential worker during pandemic	
Oneself and/or household member	86 (32%)
Someone in household	53 (19%)
No one in household is an essential worker	134 (49%)
Tested oneself for COVID-19 during pandemic	
Yes	242 (88%)
No/Not sure	34 (12%)
History of testing positive for COVID-19	
Yes	246 (90%)
No/not sure	27 (10%)
Hospitalized for COVID-19	
Yes	158 (58%)
No/not sure	115 (42%)
Number of co-morbidities (self)	
None	56 (21%)
One co-morbidity	127 47%)
Two or more co-morbidities	90 (33%)
Number of co-morbidities (household, other than oneself)	
None	65 (24%)
One co-morbidity	110 (40%)
Two or more co-morbidities	98 (36%)
# of CDC precautions followed	
Zero/one	31 (11%)
Two/three	42 (15%)
Four/five	69 (25%)
Six/seven	59 (22%)
Eight/nine	35 (13%)
Ten/eleven	37 (14%)
Received flu shot last season	
Yes	154 (58%)
No	112 (42%)
Have or will receive flu shot this year	
Yes	142 (52%)
No	90 (33%)
Undecided	41 (15%)
COVID-19 vaccination likelihood	
Definitely will not, very unlikely, somewhat unlikely	84 (31%)
Somewhat likely, very likely, definitely will	27 (10%)
Already partially or completely vaccinated	162 (59%)

**Table 2 vaccines-12-00787-t002:** The mean ($\bar{x}$) of all COVID-19 virus and vaccine knowledge topics before and after the educational interventions, as well as the mean difference (mean difference = mean after intervention—mean before intervention, ∆$\bar{x}$).

Knowledge Topics	Before Intervention ($\bar{x}$) ^1^	After Intervention ($\bar{x}$) ^2^	Mean Difference ($∆\bar{x}$)	Standard Deviation	t *	*p*-Value
Measures to protect against COVID-19 transmission	1.11	1.39	0.28	0.77	6.05	**<0.001**
Modes of COVID-19 spread	1.42	1.42	0.00	0.80	0.08	0.94
Vaccine effects	1.41	1.55	0.14	0.87	2.65	**0.008**
COVID-19 vaccine efficacy	1.41	1.6	0.19	0.83	3.74	**<0.001**
Post-vaccine behavior	1.36	1.48	0.13	0.83	2.49	**0.013**
COVID-19 side effects	1.33	1.4	0.07	0.82	1.49	0.138
COVID-19 vaccine development	0.75	0.95	0.19	0.89	3.61	**<0.001**

^1^ Mean values that show a value closer to 0 indicate an incorrect response, ^2^ indicates a partially correct response, and 2 indicates a completely correct response. * df = 272.

**Table 3 vaccines-12-00787-t003:** The mean ($\bar{x}$) of significant COVID-19 virus and vaccine attitude topics before and after the educational interventions, as well as the mean difference (mean difference = mean after intervention—mean before intervention, ∆$\bar{x}$).

Attitude Topics	Before Intervention ($\bar{x}$) ^1^	After Intervention ($\bar{x}$) ^2^	Mean Difference ($∆\bar{x}$)	Standard Deviation	t *	*p*-Value
Trust in COVID-19 vaccine effectiveness and safety	1.12	1.28	0.17	0.77	3.46	**<0.001**
Personal belief in the benefits of COVID-19 vaccination	1.29	1.37	0.09	0.55	2.517	**0.012**
Confidence in COVID-19 vaccine testing and results	1.26	1.36	0.10	0.61	2.539	**0.012**
Perception of rapid COVID-19 vaccine development	1.09	0.98	−0.10	0.69	−2.418	**0.016**
Concerns about COVID-19 vaccine side effects	1.11	0.98	−0.13	0.81	−2.591	**0.01**
Concerns about long-term effects of COVID-19 vaccines	1.22	1.06	−0.16	0.74	−3.453	**<0.001**
Challenges in discerning trustworthy vaccine information	1.13	1.04	−0.10	0.82	−1.911	0.057
Influence of trusted sources on vaccination	1.31	1.45	0.13	0.76	2.794	**0.006**
Concerns about missing work due to vaccine side effects	0.59	0.51	−0.09	0.71	−1.955	0.052

^1^ Mean values that show a value closer to 0 indicate “disagree” and ^2^ indicates “agree”. * df = 253–259.

**Table 4 vaccines-12-00787-t004:** Mean difference (∆$\bar{x}$) of significant virus and vaccine knowledge and attitude topics based on the participant’s vaccine status.

	Vaccine Status				
	Unvaccinated, Hesitated ^1^ (∆$\bar{x}$, N = 84)	Unvaccinated, Not Hesitant ^2^ (∆$\bar{x}$, N = 27)	Vaccinated (∆$\bar{x}$, N = 162)	F	*p*-Value
Knowledge topics					
Measures to protect against COVID-19 transmission	0.43	0.11	0.26	2.17	0.116
Vaccine effects	0.39	−0.11	0.04	5.75	**0.004**
COVID-19 vaccine efficacy	0.49	0.11	0.02	9.25	**<0.001**
Post-vaccine behavior	0.25	0.04	0.08	1.35	0.261
COVID-19 vaccine development	0.27	0.44	0.11	2.10	0.125
Attitude topics					
Trust in COVID-19 vaccine effectiveness and safety	0.16	0.08	0.19	0.20	0.821
Personal belief in the benefits of COVID-19 vaccination	0.21	0.21	−0.01	4.59	**0.011**
Confidence in COVID-19 vaccine testing and results	0.03	0.08	0.14	0.82	0.442
Perception of rapid COVID-19 vaccine development	−0.01	−0.12	−0.18	1.50	0.226
Concerns about COVID-19 vaccine side effects	−0.10	0.17	−0.15	1.61	0.203
Concerns about long-term effects of COVID-19 vaccines	−0.04	−0.29	−0.17	1.42	0.244
Influence of trusted sources on vaccination	0.23	−0.08	0.13	1.61	0.203
Concerns about missing work due to vaccine side effects	−0.10	−0.24	−0.05	0.78	0.462

^1^ The “Unvaccinated, hesitated” individuals include those who selected “Definitely will not”, “Very unlikely”, and “Somewhat unlikely” to the question “If given the opportunity to take a COVID-19 vaccine, how likely is it that you would get the vaccine/shot?”. ^2^ The “Unvaccinated, not hesitant” individuals include those who selected “Somewhat likely”, “Very likely”, and “Definitely will”; the “vaccinated” individuals include those who received one or more COVID-19 vaccinations.

## Data Availability

The data presented in this study are available on request from the corresponding author. The data are not publicly available due to privacy or ethical restrictions.

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
