# Peer review of "Evaluating Two Educational Interventions for Enhancing COVID-19 Knowledge and Attitudes in a Sample American Indian/Alaska Native Population"

_vaccines, 2024, doi:10.3390/vaccines12070787_

Round 1

Reviewer 1 Report

Comments and Suggestions for Authors

This is a well written, important paper, it should be published, and it fits the "Special Issue".

The use of a mobile phone application defines a population, how many of the participants opted for this option, and did this influence the results for this sub-group?

How did the authors assure cultural acceptance of the questionnaire, and how was the questionnaire designed?

How was randomization in the waiting room done?

The tables are too detailed, the average reader is lost in the scientifically sound details. Some of the tables should be given in the supplement, and described in the text.

Author Response

Dear Reviewer,

Thank you for taking the time to review our paper. Please find our responses to your comments below:

Comment 1: The use of a mobile phone application defines a population, how many of the participants opted for this option, and did this influence the results for this sub-group?

Response: Participants used either their own mobile smartphones or a research-issued iPad, both employing a QR code. Consequently, all participants used technology to complete the survey, eliminating potential bias between technology-based and paper-based surveys. We did not track specific opt-in rates for each device since all participants used some form of technology, and therefore, we did not analyze potential subgroup effects related to this variable.

Comment 2: How did the authors assure cultural acceptance of the questionnaire, and how was the questionnaire designed?

Response: We utilized the questionnaire from the study by Takagi et al. (2023). Before distribution, the study and survey were reviewed and approved by the Tribal council. The details are cited in our manuscript (see line 103), and the questionnaire is included in the supplemental material.

Citation: M. A. Takagi et al., "The impact of educational interventions on COVID-19 and vaccination attitudes among patients in Michigan: A prospective study," Front Public Health, vol. 11, p. 1144659, 2023, doi: 10.3389/fpubh.2023.1144659.

Comment 3: How was randomization in the waiting room done?

Response: The educational interventions were randomized using the Qualtrics randomization function, as described in line 112 of the manuscript.

Comment 4: The tables are too detailed; the average reader is lost in the scientifically sound details. Some of the tables should be given in the supplement and described in the text.

Response: We agree with your assessment. Tables 3 and 4 have been relocated to the supplemental material section.

Thank you once again for your valuable feedback. We have incorporated these revisions as suggested.

Reviewer 2 Report

Comments and Suggestions for Authors

Thank you for the opportunity to review this manuscript. Takagi etal addressed the COVID-19 vaccine hesitancy among a minority population of American Indian/Alaska Native through educational intervention. This is very interesting approach. However, the manuscript needs major revision. I provided some comments and questions for the authors that make their paper more suitable for publish.

The major point I have is with the survey development. The author mentioned that the survey is a validated survey. However, the donot provide a reference, neither they validated their survey for reliability and validity. This is very important. Author needs to show that the questions do address the attitude of the participants. Also, need to use some statistics such as Cronbach’s alpha (for scaled questions).

Another major limitation is excluding the non-English speakers from participating. Does this indicate that your cohort is only individual with higher education and does not represent all native population? I am not sure how this point can be addressed!

The final participated individual is 273 out of 4,123 active patients, representing 6.6% of your population. Your claim that your cohort represents 54% is not correct (you should not calculate the rate from the number of individuals you asked to participate, it should be from total population). Please correct this and add this to your limitation.

Please include the survey items in the Supplementary file.

Please fix Table 3.

Lines 183 and 213 refer to Table A1 and Table A3. In the Supplementary, there are Table A1A and Table A1B. Similarly, Table A3A and Table A3B. Which one do you mean? If you mean both, mention them in details.

Table 5 is very poorly presented. Please re-structure to be easier understood.  

Author Response

Dear Reviewer,

Thank you for your detailed feedback on our manuscript. Please find our responses to each of your comments below:

Comment 1: The major point I have is with the survey development. The author mentioned that the survey is a validated survey. However, they do not provide a reference, nor have they validated their survey for reliability and validity. This is very important. Authors need to show that the questions do address the attitude of the participants. Also, they need to use some statistics such as Cronbach’s alpha (for scaled questions).

Response: We utilized the validated questionnaire from Takagi et al. (2023) study, as referenced in our manuscript (see line 103), and have included our survey in the supplemental section. Regarding the reliability and validity of our survey, our knowledge and attitude items were loosely correlated and not strong enough for creating reliable scale scores. Our focus was on examining changes in item content in response to our intervention rather than aiming for scale scores.

Comment 2: Another major limitation is excluding non-English speakers from participating. Does this indicate that your cohort is only individuals with higher education and does not represent all native population? I am not sure how this point can be addressed!

Response: Excluding non-English speakers does not imply exclusivity to individuals with higher education. A significant portion of our target population speaks English, and clinical interactions at our facility are conducted in English. The survey was designed to be comprehensible to adults with an average reading level. In addition, the survey, the intervention and the language used were reviewed by the Tribal Council and approved as such.

Comment 3: The final participation rate is 273 out of 4,123 active patients, representing 6.6% of your population. Your claim that your cohort represents 54% is not correct (you should not calculate the rate from the number of individuals you asked to participate, it should be from the total population). Please correct this and add this to your limitations.

Response: We have clarified this point by including both the percentage of invited participants and the 6.6% representation of our study population in line 150.

Comment 4: Please include the survey items in the Supplementary file.

Response: The survey items have been included in the supplemental file, as referenced in Line 103-104.

Comment 5: Please fix Table 3.

Response: We have moved Table 3 into the Appendix (now Table A1) to enhance the clarity of our statistical analysis.

Comment 6: Lines 183 and 213 refer to Table A1 and Table A3. In the Supplementary, there are Table A1A and Table A1B. Similarly, Table A3A and Table A3B. Which one do you mean? If you mean both, please clarify in detail.

Response: We have specified the references to Tables A1, A2A-B, A4A-B, etc. accordingly.

Comment 7: Table 5 is very poorly presented. Please re-structure to be easier understood.

Response: Table 5 has been moved to the Appendix (now Table A4A-B) to improve the clarity of our statistical analysis. Additionally, we have split the table into two pages for easier comprehension.

Thank you again for your insightful comments. We have made the necessary revisions as suggested.

Reviewer 3 Report

Comments and Suggestions for Authors

The authors have submitted evidence of an intervention with an important audience which appears to have worked in the sense of increasing key knowledge and influencing reports of key perceptions related to vaccination acceptance. The question for the authors to consider is how the manuscript could be optimized as a contribution for the journal. What is generalizable in the evidence presented and what does this paper tell us about how future interventions should be developed (rather than just documenting that some interventions potentially could work)?

One of the limitations of the study lies in its pre-post design. That design suggests differences in participants' tendency to respond to knowledge and perception items in certain ways but to what extent did having the pre-test first condition people to accept the intervention in a particular way? Ideally, the authors would have included a no pre-test control but that is beyond the scope of what they did and so they should at least acknowledge that limitation further. 

We also do not know what might have happened either had the participants seen other less well-designed educational material or no material at all. The authors begin to see some evidence that material content or format mattered with their noted difference between video and printed material. It might be useful to add discussion -- in addition to the current speculation -- which would suggest steps future researchers to take to determine if it was content or format which made the difference here. 

The authors also seem to have focused on knowledge and perceptions solely as outcomes. Why wasn't future behavioral intention measured? That would be worthwhile to address in discussion.

The authors also miss an opportunity to contextualize this paper in the wider literature on health disparity interventions and health communication literature. It would be useful to connect this paper with existing literature on health equity and health information, e.g., Calac, A. J., & Southwell, B. G. (2022). How misinformation research can mask relationship gaps that undermine public health response. American Journal of Health Promotion, 36(3), 561-563. Also relevant would be Southwell, B.G., Machuca, J. O., Cherry, S. T., Burnside, M., & Barrett, N. J. (2023). Health misinformation exposure and health disparities: Observations and opportunities. Annual Review of Public Health, 44, 113-130.

If the authors can enhance their discussion of these dimensions they will improve the manuscript. 

Author Response

Dear Reviewer,

Thank you for your detailed feedback on our manuscript. Please find our responses to each of your comments below:

Comment 1: What is generalizable in the evidence presented and what does this paper tell us about how future interventions should be developed (rather than just documenting that some interventions potentially could work)?

Response: We have fortunately distributed these surveys to the wider state of Michigan in Takagi et al. (2023) and yielded similar results when it came to increasing knowledge with an educational intervention. We have added this to our discussion section under lines 247-249 “Similar to the results of this study, Takagi et al. (2023) demonstrated how the same educational interventions increased knowledge amongst patients across 18 Michigan counties [20]” This American Indian population has been included separately in this manuscript as requested by the Tribal council.

Citation: M. A. Takagi et al., "The impact of educational interventions on COVID-19 and vaccination attitudes among patients in Michigan: A prospective study," Front Public Health, vol. 11, p. 1144659, 2023, doi: 10.3389/fpubh.2023.1144659.

Comment 2: One of the limitations of the study lies in its pre-post design. That design suggests differences in participants' tendency to respond to knowledge and perception items in certain ways but to what extent did having the pre-test first condition people to accept the intervention in a particular way? Ideally, the authors would have included a no pre-test control but that is beyond the scope of what they did and so they should at least acknowledge that limitation further.

Response: We appreciate your insights into the limitations of our study design. The pre-post nature of our study does indeed introduce the possibility of participants' responses being influenced by the initial pre-test, potentially affecting their perception of the intervention. While including a no pre-test control group was beyond the scope of this study, we acknowledge this as a limitation under lines 406-412, “Considering the pre-post design of our study, it is important to note that the initial pre-test may have conditioned participants’ responses to the intervention, influencing their subsequent perceptions and attitudes. While including a no pre-test control group was beyond the scope of this study, future research could benefit from incorporating such controls to better elucidate the true impact of the intervention. This would help differentiate between the effects of the intervention itself and any potential biases introduced by the study design.”

Comment 3: We also do not know what might have happened either had the participants seen other less well-designed educational material or no material at all. The authors begin to see some evidence that material content or format mattered with their noted difference between video and printed material. It might be useful to add discussion -- in addition to the current speculation -- which would suggest steps future researchers to take to determine if it was content or format which made the difference here.

Response: That is an excellent point and something that we would love people to research in detail. We have added your points into our conclusion section, lines 435-440: “While these results highlight the potential of different modalities, further investigation is needed to determine whether the observed outcomes were primarily influenced by the content or format of the educational materials. Future studies could explore this by comparing various types of educational content, including less well-designed materials or no intervention at all, to better understand optimal strategies for promoting health literacy and behavior change within these communities.”

Comment 4: The authors also seem to have focused on knowledge and perceptions solely as outcomes. Why wasn't future behavioral intention measured? That would be worthwhile to address in discussion.

Response: We appreciate the reviewer's observation regarding the measurement of future behavioral intention in our study. While our primary focus was on assessing changes in COVID-19 knowledge and attitudes, we did include a question about the likelihood of participants getting vaccinated in the future as a measure of behavioral intention. The results of this measure are detailed in Table A6 of the supplementary materials. However, it is noteworthy that we did not observe a statistically significant change in participants' reported likelihood of getting vaccinated post-intervention.

We have included these points in lines 284-291: “In assessing the impact of educational interventions, future research could benefit from incorporating measures of behavioral intention alongside knowledge and perception outcomes. While this study focused primarily on knowledge and attitude changes, the inclusion of behavioral intention measures, such as engagement in disease prevention activities, could provide deeper insights into the practical impacts of these interventions on health behaviors within AI/AN populations. This comprehensive approach would help elucidate how educational efforts translate into actionable health decisions and contribute to ongoing efforts to improve vaccination rates and reduce health disparities among AI/AN communities.”

Comment 5: The authors also miss an opportunity to contextualize this paper in the wider literature on health disparity interventions and health communication literature. It would be useful to connect this paper with existing literature on health equity and health information, e.g., Calac, A. J., & Southwell, B. G. (2022). How misinformation research can mask relationship gaps that undermine public health response. American Journal of Health Promotion, 36(3), 561-563. Also relevant would be Southwell, B.G., Machuca, J. O., Cherry, S. T., Burnside, M., & Barrett, N. J. (2023). Health misinformation exposure and health disparities: Observations and opportunities. Annual Review of Public Health, 44, 113-130.

Response: These are good points. We agree. This is important for context. We added this to our introduction, “Current literature highlights how misinformation can exacerbate health disparities by complicating decision-making processes and undermining trust in healthcare institutions. In addition, additional studies emphasize the importance of building and maintaining trust between patients and healthcare providers as a critical strategy to mitigate the impact of misinformation and promote health equity [18, 19]. These insights demonstrate the complexities of health communication in addressing public health challenges and emphasize the need for culturally responsive interventions that foster trust and engagement within AI/AN communities. By integrating these perspectives, this study not only seeks to advance knowledge on the COVID-19 virus and vaccines among AI/AN, but also informs strategies to enhance healthcare delivery and promote health equity in marginalized populations.”

Thank you again for your insightful comments. We have made the necessary revisions as suggested.

Reviewer 4 Report

Comments and Suggestions for Authors

I appreciate your equity concern and the perspective of potential solutions, in particular education/information strategies.

Also i appreciate the pre/post longitudinal approach with clear description of methods used and results.

But, as the authors recognized it, the limitations on the selection of participants are serious : At the end, those who participated are a quite restrictive portion of the Indian population to be vaccinated: participants surveyed are those who did not dropped out, who accepted to participate, who were invited based on inclusion criteria , who were in the waiting room of a clinic. A research on "Effectiveness" " would require a very different study design, with a relevant population representation.

Author Response

Thank you for your thoughtful feedback on our manuscript. We appreciate your recognition of the strengths in our study design and methodology, particularly the longitudinal approach and clear presentation of methods and results.

We acknowledge your concerns regarding the limitations in participant selection. The constraints you highlighted, such as the restricted portion of the Indian population sampled, indeed affect the generalizability of our findings to the rest of the AI/AN populations. These limitations stemmed from practical considerations during recruitment, including adherence to inclusion criteria and logistics involved in clinic-based sampling. While our study was not designed to assess broad effectiveness due to these constraints, we understand the importance of population representation in such research. In future work, we plan to address these limitations by exploring alternative recruitment strategies to achieve a more diverse and representative sample. Another point to consider is that our recruitment was limited to access to AI/AN population we were given as defined by the Tribal Councils. In this case we were given access to patients in clinical settings. All of the above are addressed in the limitations section of the manuscript.  

We appreciate your feedback and will revise the manuscript to clarify these points. Please let us know if you have any further suggestions or concerns.

Round 2

Reviewer 3 Report

Comments and Suggestions for Authors

The authors have addressed reviewer comments and have improved the paper. 

Author Response

Thank you for your feedback and for recognizing the improvements we made based on your comments. We appreciate your time and insights, which have helped enhance our manuscript.

Reviewer 4 Report

Comments and Suggestions for Authors

My previous comments remain valid.

The design of this study cannot meet the announced objectives even as mentionned in the title.

The population sample does not fit the "population representation "criteria.

it is limited to a very selective population group: clinic visitors from a patient population living around.

Conclusions indeed could be also limited in terms of education, information strategies against vaccine hesitancy.

Author Response

Dear Reviewer,

Thank you for your detailed feedback. We appreciate your continued engagement and the opportunity to address your concerns. Here is our response to your points:

Comment 1: “The design of this study cannot meet the announced objectives even as mentionned in the title.”

Response: We appreciate your concern regarding the alignment of the study's design with its objectives. We have also changed the manuscript title to reflect our objective more clearly, “Evaluating Two Educational Interventions for Enhancing COVID-19 Knowledge and Attitudes in a Sample American Indian/Alaska Native Population” and this is also reflected in our abstract in lines 18-21, “This study conducted at a single Indian Health Service (IHS) clinic in central Michigan evaluates two educational interventions in enhancing COVID-19 knowledge and attitudes in a sample AI/AN population”. We would like to clarify how our study design effectively addresses the announced objectives:

  1. Objective Clarification:
  • The primary objective of our study is to understand and mitigate COVID-19 vaccine hesitancy among AI/AN populations through targeted educational interventions.
  • A key aspect of achieving this objective is increasing knowledge about COVID-19 and vaccines within these communities.
  1. Design and Methods:
  • Educational Interventions: We implemented tailored educational interventions approved by the Tribal council that provided accurate information about COVID-19 and vaccines, addressing specific myths and concerns prevalent in AI/AN communities.
  • Pre- and Post-Intervention Surveys: To measure the impact of our interventions, we conducted surveys before and after the educational sessions. These surveys assessed participants' knowledge, attitudes, and beliefs about the COVID-19 vaccine.
  1. Results:
  • Increased Knowledge: Our findings demonstrated a significant increase in participants' knowledge about the COVID-19 virus and vaccines following the educational interventions.
  • Shift in Attitudes: There was also a notable positive shift in attitudes toward vaccination, indicating reduced vaccine hesitancy.
  1. Meeting the Objectives:
  • The increase in knowledge directly contributes to our objective of addressing vaccine hesitancy, as informed individuals are more likely to make decisions that favor vaccination.
  • By dispelling myths and providing reliable information, the educational interventions targeted the root causes of hesitancy.
  1. Conclusion:
  • Our study design, through its focus on educational interventions and measurement of knowledge and attitude changes, effectively meets the announced objectives. The data supports that informed education is a key strategy in reducing vaccine hesitancy and promoting health equity among AI/AN populations.

Comment 2: “The population sample does not fit the "population representation "criteria.

it is limited to a very selective population group: clinic visitors from a patient population living around.”

Response: We have acknowledged the limitation of our sample population in the manuscript's limitation section, specifically noting that the study population consisted primarily of patients from a single Indian Health Service (IHS) center, as discussed in lines 423-425. We have added more details about the sample size limitations under lines 424-426 “Although efforts were made to ensure demographic diversity, the study population consisted primarily of patients from a single IHS center who did not drop out, accepted participation, met the inclusion criteria, and were in the clinic's waiting room, potentially introducing selection bias and limiting generalizability”. This introduces selection bias and limits generalizability. The primary objective of our study was to address COVID-19 vaccine hesitancy within the American Indian/Alaska Native (AI/AN) community. The selected sample of clinic visitors represents a relevant subset of this population, allowing us to implement and evaluate educational interventions in a real-world setting. The clinic was chosen due to its accessibility and the established trust it has within the community, which was crucial for the success of our intervention, factors which are important in successful interventions as discussed in previous studies which we discussed in lines 266-283. While our study provides valuable insights, we recognize the importance of broader representation. Future research should aim to include a more diverse AI/AN population by reaching out to multiple IHS centers, incorporating participants from various geographical locations, and including individuals who face barriers to clinic access. Despite the noted limitations, our study addresses significant gaps in the existing research, such as the prospective assessment of educational intervention effects and the comparison of different educational modalities within the AI/AN community.

Comment 3: “Conclusions indeed could be also limited in terms of education, information strategies against vaccine hesitancy.”

Response: We acknowledge that our study, conducted at a single IHS clinic in central Michigan, has limitations in terms of generalizability and the range of educational strategies examined. These limitations were discussed in the manuscript, particularly noting the need for further investigation into the content and format of educational materials and potential barriers to their effectiveness.

Our study highlights the critical role of educational interventions in addressing vaccine hesitancy among AI/AN populations. Specifically, it demonstrates the comparable effectiveness of both video and infographic interventions in increasing COVID-19 knowledge. These findings suggest that different modalities can be beneficial, providing a basis for future research to explore optimal strategies for promoting health literacy and behavior change.

We agree that further research is needed to:

  • Compare various types of educational content, including less well-designed materials or no intervention, to understand the influence of content and format.
  • Address barriers associated with different intervention modalities, such as hearing or vision impairments for videos and reading impairments for infographics.
  • Integrate cultural components and consider the role of native language speakers and Traditional Knowledge Holders in educational initiatives.

Our conclusion underscores the historical underfunding of IHS and the urgent need for increased financial support for tailored interventions. Investing in educational initiatives and culturally sensitive approaches can help address healthcare disparities and promote health equity within AI/AN communities.

While our study provides valuable insights into the effectiveness of educational interventions, we recognize that it represents an initial step. The conclusions drawn are based on the observed data and highlight the importance of continued research and investment in this area. We suggest that future studies should expand on our findings by incorporating a broader range of educational strategies and addressing the identified limitations to enhance the overall understanding of effective interventions against vaccine hesitancy.

We hope this response clarifies the contributions of our study and the areas for future research. We appreciate your insights and suggestions to further improve our work.